# Systematic design of pulse dosing to eradicate persister bacteria

**Garima Singh, Mehmet A. Orman, Jacinta C. Conrad, Michael Nikolaou**  *

Chemical and Biomolecular Engineering Department, University of Houston, Houston, Texas, United States of America

* nikolaou@uh.edu

**Data Availability Statement:** All code and data are public at Github: https://github.com/Nikolaou/Systematic-Design-of-Pulse-Dosing-to-Eradicate-Persister-Bacteria.

## Abstract

A small fraction of infectious bacteria use persistence as a strategy to survive exposure to antibiotics. Periodic pulse dosing of antibiotics has long been considered a potentially effective strategy towards eradication of persisters. Recent studies have demonstrated through *in vitro* experiments that it is indeed feasible to achieve such effectiveness. However, systematic design of periodic pulse dosing regimens to treat persisters is currently lacking. Here we rigorously develop a methodology for the systematic design of optimal periodic pulse dosing strategies for rapid eradication of persisters. A key outcome of the theoretical analysis, on which the proposed methodology is based, is that bactericidal effectiveness of periodic pulse dosing depends mainly on the ratio of durations of the corresponding on and off parts of the pulse. Simple formulas for critical and optimal values of this ratio are derived. The proposed methodology is supported by computer simulations and *in vitro* experiments.

## Author summary

Administering antibiotics in periodic pulses that alternate between high and low concentration has long been known as a possible dosing strategy to treat stubborn infections caused by bacteria known as *persisters*. Such bacteria use clever mechanisms to survive otherwise lethal temporary exposure to antibiotics and to resume normal activity upon antibiotic removal. Persisters pose a serious health problem. Recent studies have elucidated mechanisms of persistence and have confirmed that pulse dosing, if designed appropriately, can indeed be effective. However, effective pulse dosing design has been mainly handled by trial and error, requiring relatively extensive experimentation. Here we develop a method for rapid systematic design of effective pulse dosing. The method relies on a simple mathematical model and a minimal amount of standard experimental data. We derive corresponding design formulas that explicitly characterize the shape of generally effective or optimal periodic pulses. We tested our method through computer simulations and *in vitro* experiments, as well as on prior literature data. In all cases, the outcomes on persister bacteria eradication predicted by our method were confirmed. These results pave the way for ultimately developing effective pulse dosing regimens in realistic situations *in vivo*.

**Funding:** The Institute of Allergy and Infectious Diseases of the National Institutes of Health under award number 371 R01AI140287 (to MN) partially supported the research reported in this publication. The University of Houston under GEAR grant 2091-H0067-B0421-I113494 (ST 61553) (to JCC) partially supported this study. The content is solely the responsibility of the authors and does not necessarily represent the official views of the National Institutes of Health. The funders had no role in study design, data collection and analysis, decision to publish, or preparation of the manuscript.

**Competing interests:** The authors have declared that no competing interests exist.

# 1 Introduction

Persister cells are a small fraction of a bacterial population in a physiological state that enables them to survive otherwise lethal doses of antibiotics. While these cells remain in the state of persistence, they cannot be killed by conventional antibiotics, unless the cells phenotypically switch to the normal cell state and become susceptible to antibiotics again [1]. Persisters are enriched in biofilms [2] and implicated in many chronic infections such as tuberculosis and relapse of infections such as recurrent urinary tract infection or cystic fibrosis [3–8]. Unlike antibiotic-resistant mutant cells, persisters are phenotypic variants that survive treatments without acquiring heritable genetic changes [9]. However, prolonged persistence creates favorable conditions for the emergence of the mutant cells [10,11].

Although the term *persister* was coined in the 1940s [12], our fundamental knowledge of persisters has accelerated only in the last two decades with the advent of new technologies enabling us to study cell heterogeneity [1,13]. Persisters survive via a plethora of putative molecular mechanisms [14] and recent studies have started shedding light on how diverse and multifaceted phenomena can link effects of initial states and environmental factors to phenotypic changes during persister formation, survival, and return to normal state [14–20]. In view of that complexity, developing anti-persister therapeutics remains a challenge. Such development can be broadly classified into two categories: (a) developing new anti-persister drugs, and the most common approach (b) manipulating the dosing regimen of approved antibiotics, used either individually or in combination. The former relies on detailed knowledge about persister mechanisms (formation, survival, and resuscitation) and is certainly significantly more time-consuming and resource-intensive than the latter. Within the latter category, strategies can be based on bacterial population dynamics without complete knowledge of molecular mechanisms, capitalizing instead on the back and forth switching of persisters from normal state to persistence and vice-versa. The idea of periodic pulse dosing to kill persisters is as old as the term persister itself [12]. Yet, there are only sporadic studies on pulse dosing, examining *in vitro* efficacy [21, 22] and model fitting of experimental data or characterization of effective pulse dosing strategies [23–27]. A remaining challenge is a simple systematic design of an effective periodic pulse dosing regimen, comprising periods of antibiotic administration at high and low concentrations successively. Indeed, the alternating periods of antibiotic application (on) and removal (off) are critical to the success of pulse dosing strategies [22,28], as use of an inappropriate strategy fails to achieve eradication [29, 30]. Attempts to connect experiments and modelling [28] have underscored the importance of characterizing optimal dosing regimens in a simple quantitative fashion.

The present study aims at addressing this issue. The specific contributions of this study are (a) rigorous theoretical justification that the efficacy of pulse dosing with alternating on/off periods of antibiotic administration depends on the ratio of the corresponding on/off periods of a pulse rather than on their individual values; (b) explicit formulas for robustly optimal values of this ratio in terms of easily estimated parameters; and (c) experimental confirmation *in vitro* of both positive and negative model predictions (bacterial eradication or not, respectively). In the rest of the paper, we describe our experimental and modeling studies, present our main results, and close with suggestions for future studies. Proofs and details, to the extent that they provide insight, are included in S1 Text.

# 2 Materials and methods

## 2.1 Experimental

**2.1.1 Bacterial strain and plasmid.** *Escherichia coli* wild-type (WT) with a pQE-80L plasmid encoding a green fluorescent protein (GFP) was used. These were obtained from Dr. Mark P. Brynildsen of Princeton University.

**2.1.2 Media and chemicals.** Luria-Bertani (LB) broth was used for all liquid cultures. LB broth was prepared by dissolving its components (10g Tryptone, 10 g Sodium Chloride, 5 g Yeast Extract) in 1 L distilled water and sterilized with an autoclave. Ampicillin (Sigma Aldrich) was used to treat cells at a constant dose of 100 $\mu$g/mL. Phosphate Buffered Saline (PBS) was used to wash the cells to remove Ampicillin. LB agar medium was prepared by dissolving 40 g LB agar premix in 1 L DI water and sterilized with an autoclave. LB agar medium was used to enumerate colony forming units (CFUs) of *E. coli* [13,31,32].

**2.1.3 Constant (control) and pulse dosing experiments.** Bacteria were exposed to Ampicillin in two ways: at constant Ampicillin concentration (control) and at pulsed antibiotic concentration of the same amplitude (pulse on/off dosing, Fig 1).

Each pulse experiment was started by inoculating (1,100-fold) an overnight (24 h) culture of *E. coli* into 25 ml of LB. For selection and retention of plasmids in bacterial cells, 50 $\mu$g/mL kanamycin was added in culture media [32]. To induce fluorescent protein expression, 1 mM IPTG was used [32]. The overnight culture was prepared from frozen glycerol stock (-80˚C). All cells were cultured in a shaker at 37˚C and 250 rpm. The pulse dosing schedule was: (a) expose bacteria to Ampicillin (100 $\mu$g/mL) for $t_{on}$ h, and (b) wash treated cells and grow in fresh media for $t_{off}$ h. Treated cells were washed with PBS buffer solution to remove the antibiotics. Cells were serially diluted in PBS using 96-well plates, spotted on LB agar, and incubated at 37˚C for 16 h to enumerate CFUs. Bacteria population size was assessed by colony counting on LB-Agar plates.

## 2.2 Modelling and simulation

A two-state model [1], comprising two cell-balance differential equations for normal and persister cells, was used in all analysis:

$$\frac{dn}{dt} = K_n n(t) + bp(t) \qquad (1)$$

$$\frac{dp}{dt} = an(t) + K_p p(t) \qquad (2)$$

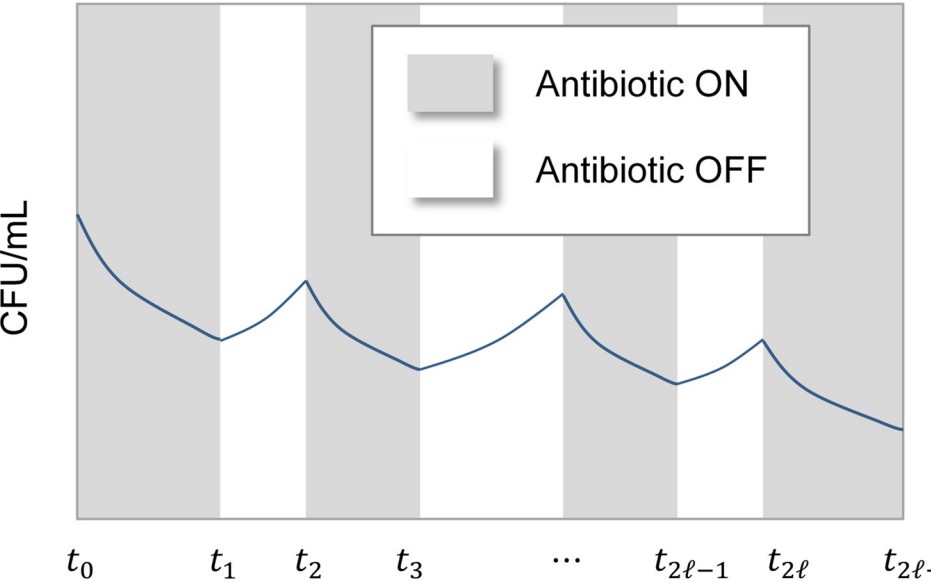

**Fig 1. A schematic of the outcome of a pulse dosing regimen showing declining peaks and dips of a bacterial cell population over time.**

or, equivalently,

$$\underbrace{\begin{bmatrix} dn/dt \\ dp/dt \end{bmatrix}}_{d\mathbf{x}/dt} = \underbrace{\begin{bmatrix} K_n & b \\ a & K_p \end{bmatrix}}_{\mathbf{A}} \underbrace{\begin{bmatrix} n(t) \\ p(t) \end{bmatrix}}_{\mathbf{x}(t)} \tag{3}$$

where

$n(t)$ = Number of normal cells at time $t$

$p(t)$ = Number of persister cells at time $t$

$$\mathbf{x}(t) \stackrel{\text{def}}{=} \begin{bmatrix} n(t) \\ p(t) \end{bmatrix}$$

$$\mathbf{A} \stackrel{\text{def}}{=} \begin{bmatrix} K_n & b \\ a & K_p \end{bmatrix}$$

$a$ = switch rate from normal to persister state

$b$ = switch rate from persister to normal state

$K_n \stackrel{\text{def}}{=} \mu_n - k_n - a$ = net decline / growth rate of normal cells

$K_p \stackrel{\text{def}}{=} \mu_p - k_p - b$ = net decline / growth rate of persister cells

$\mu_n, \mu_p$ = growth rate of normal or persister cells, respectively

$k_n, k_p$ = kill rate of normal or persister cells, respectively

The parameters $a$, $b$, $K_n$, $K_p$ in Eqs (1) and (2) are generally distinct when the antibiotic is administered or not (on/off), resulting in corresponding matrices $\mathbf{A}_{\text{on}}$, $\mathbf{A}_{\text{off}}$ in Eq (3). In the constant dosing (control) experiment the antibiotic remained always on, whereas for pulse dosing experiments the antibiotic alternated between on (administered) and off (not administered) with corresponding durations $t_{\text{on}}$, $t_{\text{off}}$. Therefore, to fit the full data set (at constant and pulse dosing) by Eqs (1) and (2), parameter estimation generally entailed eight values for estimates of $\{a, b, K_n, K_p\}_{\text{off}}$ and $\{a, b, K_n, K_p\}_{\text{on}}$ during on and off periods, respectively. In addition, because data fit relied on measurements of the total number of cells,

$$c(t) \stackrel{\text{def}}{=} n(t) + p(t) \tag{4}$$

a ninth parameter, namely the initial fraction of persister cells, $f_0$, was estimated.

Mathematica and MATLAB were used for all modelling, parameter estimation, and analysis computations.

## 3 Results

We present first a theoretical analysis for characterization of optimal pulse dosing regimens based on the model of Eqs (1) and (2), and subsequently present a series of experiments for validation of analysis results *in vitro*.

### 3.1 Design of pulse dosing for rapid bacterial population reduction

**3.1.1 Characterizing $\{t_{\text{on}}, t_{\text{off}}\}$ *for decline of a bacterial population*.**   Analysis of Eqs (1) and (2) (Appendix A in S1 Text) suggests that successive local peaks of the bacterial population size, $c(t)$, at times $t_{2\ell} \stackrel{\text{def}}{=} \ell(t_{\text{on}} + t_{\text{off}})$, as generally depicted in Fig 1, are characterized as

$$c(t_{2\ell}) = p_1 \lambda_1^\ell + p_2 \lambda_2^\ell, \ell = 0, 1, 2, \ldots \tag{5}$$

**Table 1. Parameter estimates fitting the experimental data of Fig 6.** (standard errors reported).

| Parameter | Antibiotic on | Antibiotic off |
|---|---|---|
| $K_n$ | −3.8±0.4 | 1.35±0.09 |
| $K_p$ | −0.2±0.2 | −1.2±0.4 |
| $a$ | 0 | 0 |
| $b$ | 0 | 1.2±0.4 |
| $f_0$ | 9E−07±1.6E−05 | |
| $\log_{10} c_0$ | 7.8±0.4 | |

where $\lambda_1, \lambda_2$ are the eigenvalues of the matrix

$$\mathbf{M} \stackrel{\text{def}}{=} \exp(\mathbf{A}_{\text{off}} t_{\text{off}}) \exp(\mathbf{A}_{\text{on}} t_{\text{on}}) \tag{6}$$

and $p_1, p_2$ are coefficients depending on the model parameters and initial conditions.

It can be shown (Appendix A in S1 Text) that $\lambda_1, \lambda_2$ in Eq (5) are inside the unit disk. As a result, the pattern of $c(t_{2n})$ in Eq (5) is downward if and only if the respective on/off periods of pulse dosing, $t_{\text{on}}, t_{\text{off}}$, satisfy the inequality

$$\frac{t_{\text{off}}}{t_{\text{on}}} < \left(\frac{t_{\text{off}}}{t_{\text{on}}}\right)_c \stackrel{\text{def}}{=} -\frac{K_{n,\text{on}}}{K_{n,\text{off}}} \tag{7}$$

Note that the critical value $(t_{\text{off}}/t_{\text{on}})_c$ is positive, because $K_{n,\text{on}}<0$, $K_{n,\text{off}}>0$.

As an example, for the estimates $K_{n,\text{on}}, K_{n,\text{off}}$ shown in Table 1, Eq (7) yields

$$\frac{t_{\text{off}}}{t_{\text{on}}} < 2.8 \tag{8}$$

for peak-to-peak decline, as shown in Fig 2.

**3.1.2 Characterizing {$t_{\text{on}}$, $t_{\text{off}}$} for rapid peak-to-peak decline of the bacterial population.** For pulse dosing with fixed on/off periods $t_{\text{on}}, t_{\text{off}}$ that satisfy the inequality in Eq (7), it can be shown (Appendix B in S1 Text) that Eq (3) implies that successive peaks of $c(t)$ at times $t_{2n}$ (Fig 1) decline exponentially over time at a rate, $k$ characterized as

$$k \stackrel{\text{def}}{=} \frac{\sqrt{\ln(\lambda_1)\ln(\lambda_2)}}{t_{\text{on}} + t_{\text{off}}} = \frac{\sqrt{(K_{n,\text{off}}x + K_{n,\text{on}})(K_{p,\text{off}}x + K_{p,\text{on}})}}{1 + x} \tag{9}$$

where

$$x \stackrel{\text{def}}{=} \frac{t_{\text{off}}}{t_{\text{on}}} \tag{10}$$

or, equivalently, with a time constant, $\tau = 1/k$.

It follows immediately (Appendix A in S1 Text) that the maximum peak-to-peak decline rate for $k$ (as shown in Eq (9)) is attained at

$$\left(\frac{t_{\text{off}}}{t_{\text{on}}}\right)_{\text{opt}} = \frac{2R_n R_p - R_n - R_p}{2 - R_n - R_p} \approx \frac{R_n}{R_n - 2} \tag{11}$$

where

$$R_p \stackrel{\text{def}}{=} \frac{K_{p,\text{on}}}{K_{p,\text{off}}}, \ R_n \stackrel{\text{def}}{=} \frac{K_{n,\text{on}}}{K_{n,\text{off}}} \tag{12}$$

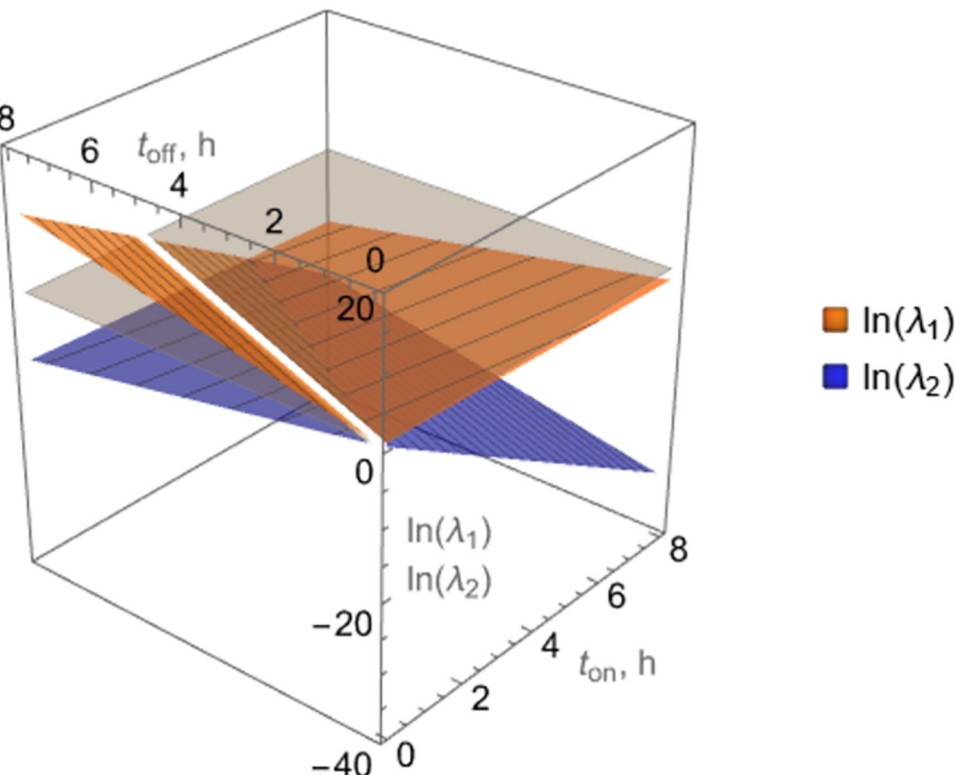

**Fig 2. Eigenvalues $\lambda_1, \lambda_2$ of the matrix $M \triangleq \exp(A_{off}t_{off})\exp(A_{on}t_{on})$ governing successive peaks of the bacterial population size, $c(t_{2n})$, under pulse dosing with successive on/off pulses of respective durations $t_{on}, t_{off}$, given parameter values in Table 1.** Values of $t_{on}, t_{off}$ that keep both $\lambda_1, \lambda_2$ below 1 (equivalently $\ln(\lambda_1), \ln(\lambda_2)$ below 0) are shown. The white line corresponding to $(t_{off}/t_{on})_c \approx 2.8$, Eq (8), on the gray plane at 0 indicates the threshold (critical) value of $t_{off}/t_{on}$, above which $\lambda_1 > 1$.

and the approximation in Eq (11) is based on

$$\frac{K_{p,\text{on}}}{K_{p,\text{off}}} \approx 0 \tag{13}$$

Note that $(t_{off}/t_{on})_{opt}$ in Eq (11) depends on the ratios $R_p \triangleq K_{p,\text{on}}/K_{p,\text{off}}, R_n \triangleq K_{n,\text{on}}/K_{n,\text{off}}$, rather than on the individual values of $K_{p,\text{on}}, K_{p,\text{off}}, K_{n,\text{on}}, K_{n,\text{off}}$. Note also that combination of Eqs (11) and (7) immediately connects the optimal and critical values of the ratio $t_{off}/t_{on}$ as

$$\left(\frac{t_{\text{off}}}{t_{\text{on}}}\right)_{\text{opt}} \approx \frac{(t_{\text{off}}/t_{\text{on}})_c}{(t_{\text{off}}/t_{\text{on}})_c + 2} \tag{14}$$

Application of Eq (11) yields a profile or the optimal ratio $(t_{off}/t_{on})_{opt}$ in terms of the ratios $R_p, R_n$ in Eq (12) as shown in Fig 3. In that figure, estimates from the experimental data of Fig 2 fitted by parameter estimates in Table 1 are used to mark the indicated point.

To visualize Eqs (7), (9), and (11), estimates of $\{K_n, K_p\}_{\text{off}}$ and $\{K_n, K_p\}_{\text{on}}$ in Table 1 were used to produce a characterization of the peak-to-peak decline rate as a function of $t_{on}, t_{off}$, as shown in Fig 4.

A remarkable implication of Eq (11) is that an optimal ratio $t_{off}/t_{on}$ (in the sense defined in this section) can be approximately assessed by mere knowledge of the ratio $K_{n,\text{on}}/K_{n,\text{off}}$, which,

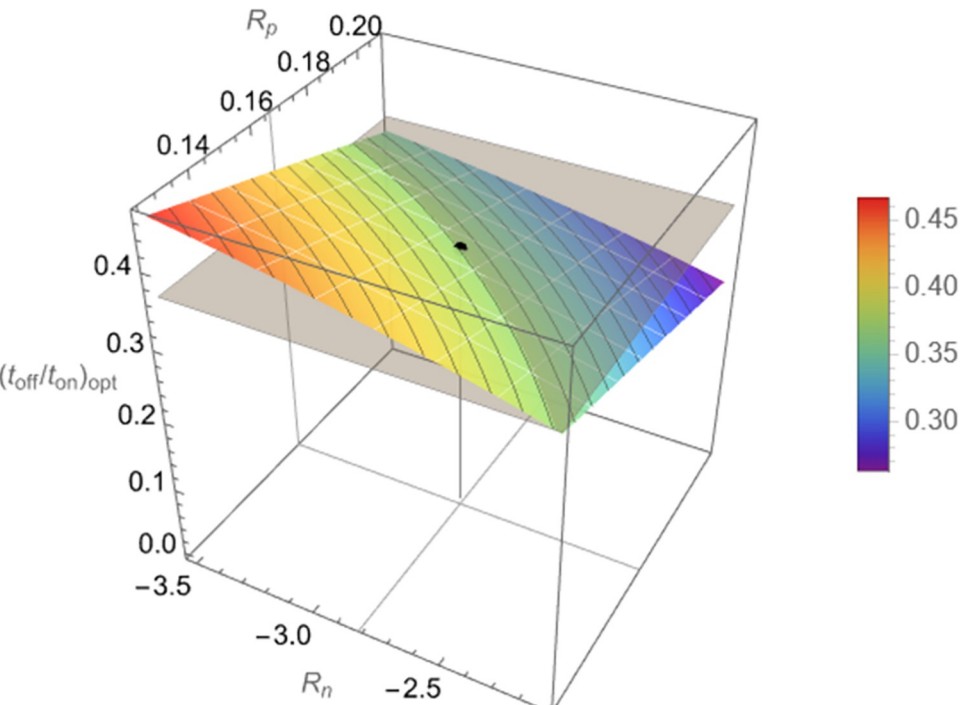

**Fig 3. Optimal values $(t_{on}/t_{off})_{opt}$ of the ratio $t_{on}/t_{off}$ that yield maximum decline rate, $k$, of successive peaks of bacterial population counts, $c(t_{2\ell})$, as a function of the quantities $R_p \cong \frac{K_{p,on}}{K_{p,off}}$, $R_n \cong \frac{K_{n,on}}{K_{n,off}}$, according to Eq (11).** Values of $R_n$, $R_p$ are considered in the range ±25% of the experimentally estimated values, which correspond to $(t_{on}/t_{off})_{opt} \approx 0.4$, marked by the black dot and gray horizontal plane through it.

in turn, can be easily obtained from two simple short-term experiments, namely standard time-growth and time-kill. The corresponding initial slopes in such experiments immediately yield $K_{n,off}$ (time-growth) and $K_{n,on}$ (time-kill), as discussed next. This is a significant simplification of the parameter estimation task for the purpose of pulse dosing regimen design, because it reduces the number of parameters to be estimated from nine (cf. section 2.2) to two. In addition, these two parameters are much easier to estimate than the remaining seven, whose accurate estimates are hard to obtain, as discussed in section 3.3.

## 3.2 Constant and pulse dosing experiments

Data from a simple time-growth experiment (no antibiotic) and a time-kill experiment (Ampicillin at constant concentration 100 μg/mL) are shown in Fig 5. The slopes from respective curve fits of the data in Fig 5 yield

$$K_{n,off}/\ln(10) \approx 0.5, K_{n,on}/\ln(10) = -1.4 \tag{15}$$

Detailed statistics are provided in Appendix C in S1 Text. Note that the pattern shown in Fig 5B indicates exponential decline (corresponding to a straight line of the bacterial population logarithm) as the almost horizontal part of the biphasic trend, which is typical of persisters [2], has not yet been reached within 3 hours. That trend is reached through continuation of the experiment after 3 hours, as indicated by the red data points and corresponding bend of the line fitted in Fig 6.

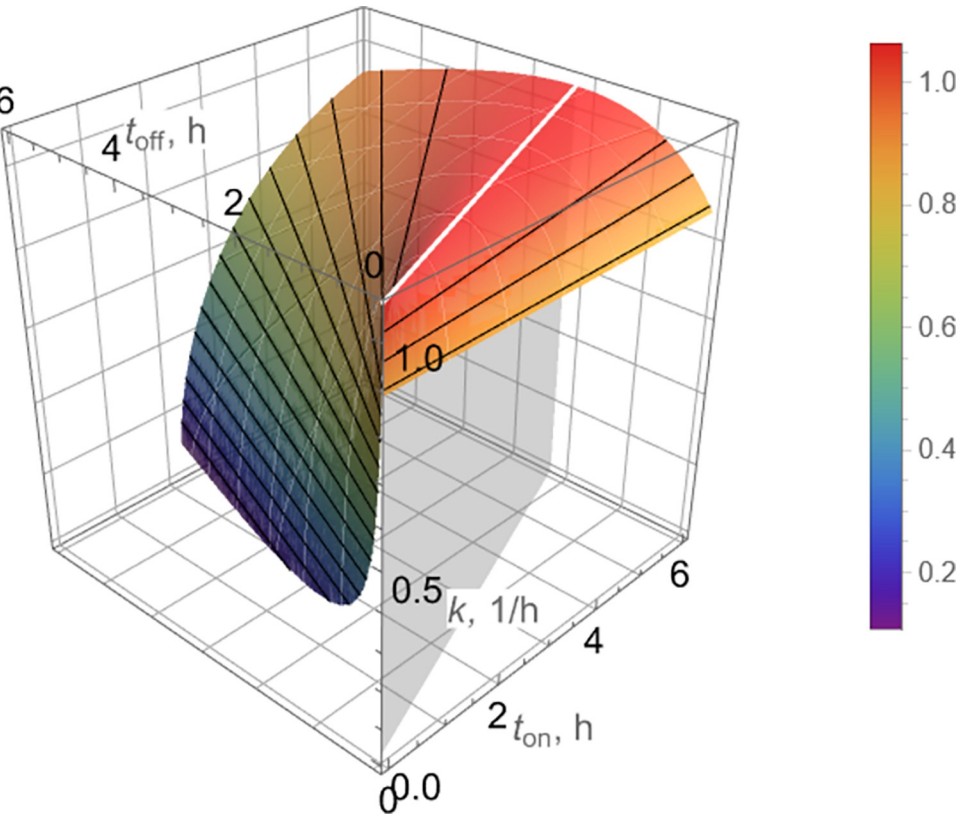

**Fig 4. Peak-to-peak decline rate, $k$, for bacterial count peaks $c(t_{2\ell})$ (Eq (9)) as a function of on/off pulse durations, $t_{on}$, $t_{off}$, respectively, for pulse dosing.** The decline rate, $k$, achieves a maximum for $(t_{off}/t_{on})_{opt} \approx 0.4$ according to Eq (11) (white line) and remains relatively flat for small variations of $t_{on}$, $t_{off}$ near $(t_{off}/t_{on})_{opt} \approx 0.4$. Note also that isolines (black) are straight, indicating dependence of $k$ on $(t_{off}/t_{on})$ rather than on individual values of $t_{on}$, $t_{off}$.

Using the above two estimates for $K_{n,off}$, $K_{n,on}$ in Eq (12) to calculate $R_n$ and substituting the resulting value of $R_n$ into Eq (11) yields

$$\left(\frac{t_{off}}{t_{on}}\right)_{opt} \approx 0.6 \tag{16}$$

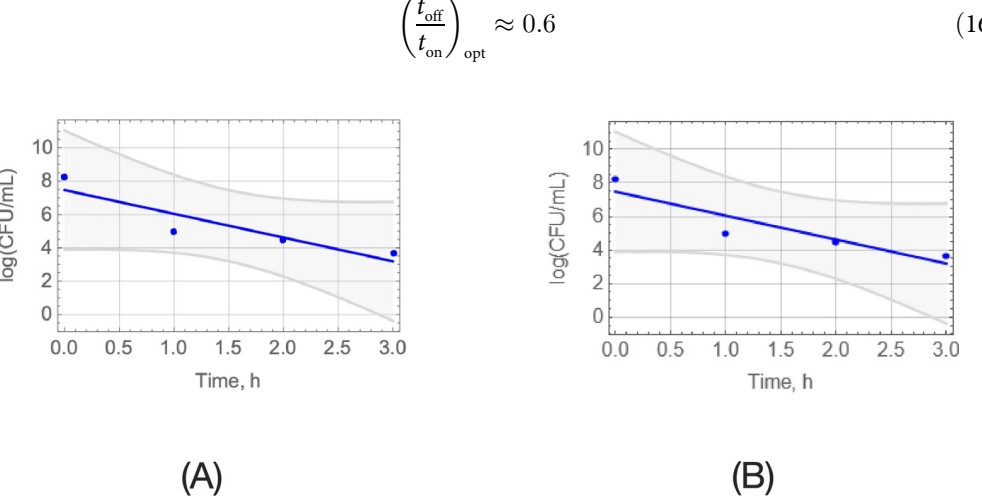

(A)                                                    (B)

**Fig 5.** Time-growth (a) and time-kill (Ampicillin at 100 μg/mL) (b) experiments for generation of data to assess $t_{on}/t_{off}$ for design of pulse dosing. 95%-confidence bands are shown. The slope of the cyan straight line (a) indicates the initial slope of the blue line and corresponds to $K_{n,off}/\ln(10)$, whereas the blue line slope (b) is $K_{n,on}/\ln(10)$.

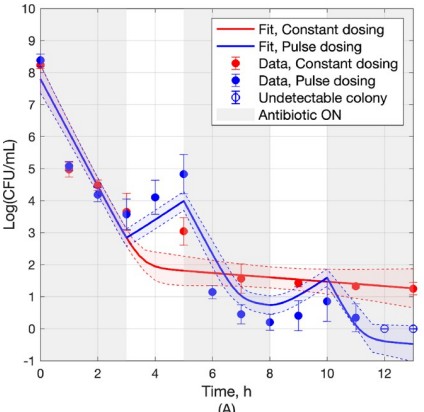
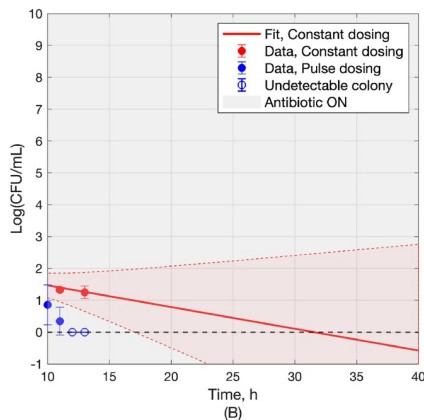

**Fig 6.** (a) Experimental data and model fit for bacterial population over time, resulting from constant dosing (red) and uniform pulse dosing at ratio $t_{\text{off}}/t_{\text{on}} = 2/3$ (blue). The hollow blue circles at 12h and 13h are set to 0 by convention, to indicate undetectable colony forming units (CFU). (b) Fitted model projection for the outcome of constant dosing to reach 0. 68%-confidence bands (1 standard error) are shown.

Therefore, to be close to the value of 0.6, pulse dosing with

$$t_{\text{on}} = 3\text{h}, t_{\text{off}} = 2\text{h} \tag{17}$$

was implemented in validation experiments *in vitro*.

The outcome of pulse dosing using the above $t_{\text{on}}$, $t_{\text{off}}$, along with additional data for continuation of the constant dosing (control) experiment to 13h, are shown in Fig 6. In the case of constant dosing, colonies were seen in all replicates at the end of the 13h treatment (5−100 CFU/mL) whereas in the case of pulse dosing no colony was observed in any replicate in the latter part of the last $t_{\text{on}}$ cycle. Note that both Figs 5 and 6 indicate a 5 log decrease of viable cells over the first 3 hours.

### 3.3 Model parameter estimation with full data set

To confirm that the model used in the analysis for development of the formulas applied in the approach proposed to pulse dosing design, the parameters in Eqs (1) and (2) were fit to the full set of experimental data of Fig 6, with corresponding curves shown in Fig 6 and parameter estimates shown in Table 1.

### 3.4 Pulse dosing design validation

Eqs (7) and (8) indicate that bacteria eradication would be achieved if the pulse dosing period ratio $t_{\text{off}}/t_{\text{on}}$ remained below 2.8, based on the short-term data fit (Fig 5), or below 2.4, based on full data fit (Fig 6). This assessment is confirmed via both simulation and experiment.

**3.4.1 Simulation.** Integration of Eqs (1) and (2) for parameter estimates based on Fig 6 yields the results of Fig 7. That figure shows profiles of the total bacterial population over time for various values of $\{t_{\text{on}}, t_{\text{off}}\}$, anticipated by Fig 4 to correspond to either eradication or growth of the population.

**3.4.2 Experiment.** In addition to the experimental data in Fig 6, which confirmed rapid bacterial eradication for pulse dosing at ratio $t_{\text{off}}/t_{\text{on}} = 2/3$, example cases of pulse dosing at ratios $t_{\text{off}}/t_{\text{on}}$ anticipated by the developed theory and by computer simulations not to result in bacterial eradication were also tested through *in vitro* experiments, as shown in Fig 8. This figure shows the outcome (dots) of two pulse dosing experiments corresponding to the same $t_{\text{off}}/t_{\text{on}} = 6$, above the

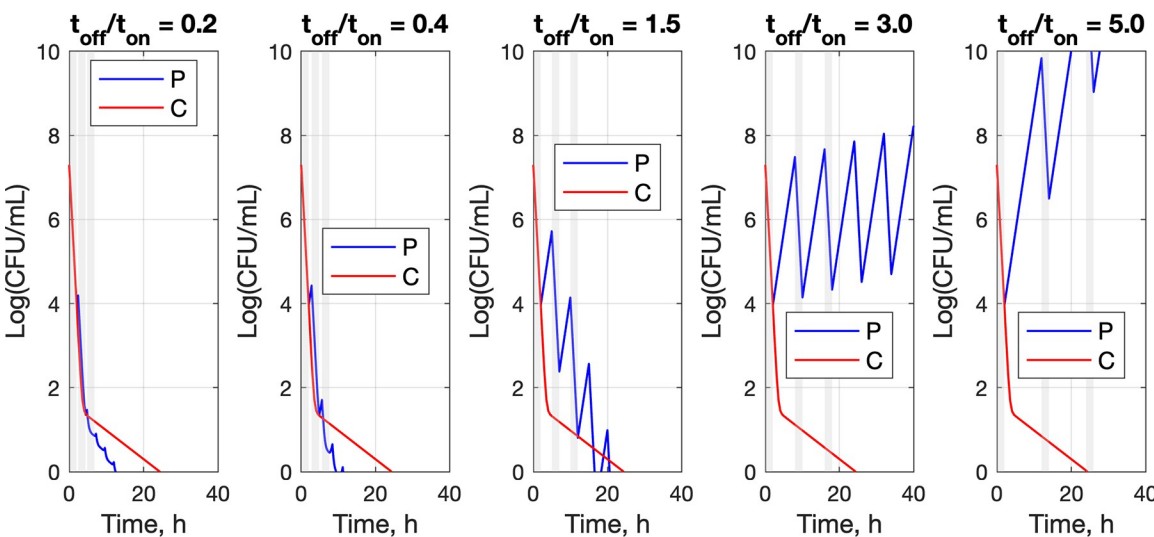

**Fig 7.** Solution of Eqs (1) and (2) for constant dosing (red) and for uniform pulse dosing (blue) with $t_{on}$ = 2h and different values of $t_{off}/t_{on}$, given estimates of $\{a, b, K_n, K_p\}_{off}$ and $\{a, b, K_n, K_p\}_{on}$ obtained from the experimental data of Fig 6. Outcomes are as anticipated by Fig 4, i.e. eradication at different rates is observed in (a)-(c) and lack of eradication in (d), (e).

critical value $(t_{off}/t_{on})_c$ = 2.8 (Eq (8)), for $t_{on}$ = 0.5h and $t_{on}$ = 3h. Model predictions (Eqs (1) and (2) with parameter estimates from Table 1 based on data of Fig 6) are also shown.

Note that for $\{t_{on}, t_{off}\}$ = {3h, 18h} the linear model of Eqs (1) and (2) correctly predicts the observed peak-to-peak upward trend, but the model crosses the bacterial population saturation limits (Eq (SI-23)) which are at about 9 log(CFU/mL) (Appendix C in S1 Text) hence the quantitative discrepancy.

Additional experiments were also conducted to test the robustness of pulse dosing in case of non-uniform perturbations of $t_{on}$ around 3h and $t_{off}$ around 2h. The results, shown in Fig 9, suggest rapid bacterial eradication (within about 10h), on par with uniform pulse dosing (cf. results shown in Figs 6 and 7).

## 4 Discussion

### 4.1 Summary of results

In summary, the presented method to pulse dosing regimen design relies on Eq (11), which expresses the optimal ratio $(t_{off}/t_{on})_{opt}$ as approximately a function of $R_n \stackrel{\text{def}}{=} K_{n,on}/K_{n,off}$, where

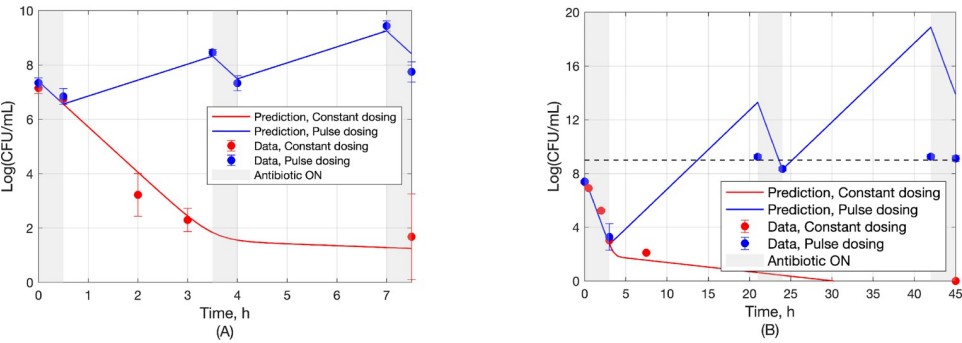

**Fig 8.** Experimental values (dots) and model predictions (lines) for constant dosing (red) and pulse dosing with $t_{off}/t_{on}$ = 6 > $(t_{off}/t_{on})_c$ (blue) for (a) $\{t_{on}, t_{off}\}$ = {0.5h, 3h} and (b) $\{t_{on}, t_{off}\}$ = {3h, 18h}.

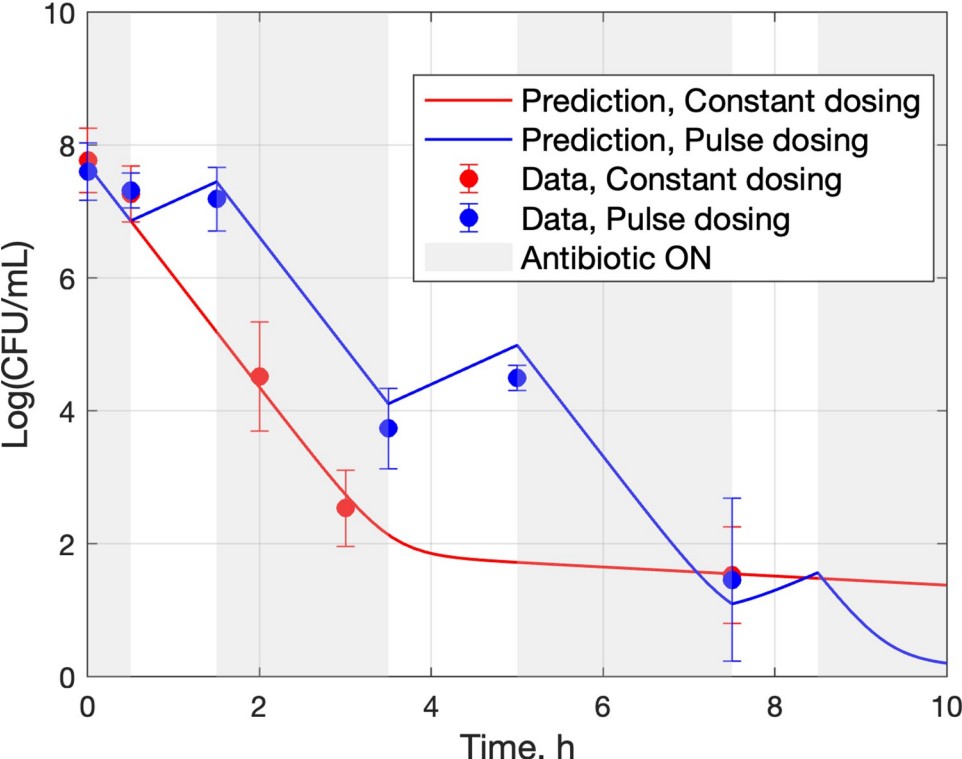

**Fig 9. Outcome of non-uniform pulse dosing (and control) for non-uniform perturbations of pulse durations around their nominal values $t_{on}$ = 3h, $t_{off}$ = 2h.**

the parameters $K_{n,off}$, $K_{n,on}$ are easy to obtain from the early parts of time-growth and time-kill experiments, respectively, as demonstrated in Fig 5. Starting with this, the efficacy of a near optimal proposed design is experimentally validated in Fig 6, the sub-optimality of designs with $t_{off}/t_{on}$ other than $(t_{off}/t_{on})_{opt}$ is illustrated in Fig 7, predictions for the complete failure of pulse dosing regimens with $t_{off}/t_{on}$ above the critical value (Eq (7)) are experimentally validated in Fig 8, and the robustness of proposed dosing regimen designs is experimentally validated in Fig 9.

Additional details are discussed in sections 4.2 and 4.3. Finally, an elucidation of literature results in view of the proposed theory is presented in section 4.4.

## 4.2 Design of pulse dosing for rapid bacterial population reduction

**4.2.1 Characterizing $\{t_{on}, t_{off}\}$ *for decline of a bacterial population*.** The analysis presented relies on the linear model of Eqs (1) and (2). That model is clearly not valid when the bacterial population reaches its saturation limits after long growth periods (e.g. Fig 5A and Eq (SI-23) in Appendix C in S1 Text). Therefore, Eq (7), which places an upper bound on the ratio $t_{on}/t_{off}$ of pulse dosing, is understood to hold for values of $t_{off}$ that do not drive the bacterial population beyond logarithmic growth to saturation.

**4.2.2 Characterizing $\{t_{on}, t_{off}\}$ *for rapid peak-to-peak decline of the bacterial population*.** Selection of the geometric average of two decline rate constants,

$$k_1 \overset{\text{def}}{=} -\frac{\ln(\lambda_1)}{(t_{on} + t_{off})}, k_2 \overset{\text{def}}{=} -\frac{\ln(\lambda_2)}{(t_{on} + t_{off})} \tag{18}$$

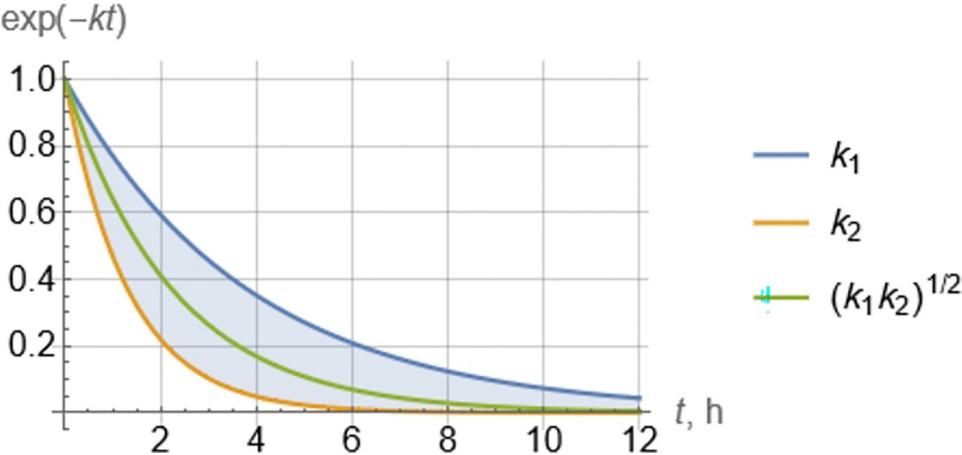

**Fig 10.** Exponential peak-to-peak decline rate modes corresponding to each individual rate $k_1$, $k_2$ (Eq (18)) and the geometric average rate $k \stackrel{\triangle}{=} (k_1 k_2)^{1/2}$ (Eq (9)).

in Eq (9), captures well an overall decline rate (Fig 10) when early parts of the decline are important.

An alternative, focusing more on the long-term decline rate, would be to choose the dominant (smaller) of the two decline rates $k_1$, $k_2$ (Fig 10). The results would be somewhat different quantitatively, but of similar nature. This is exemplified in Fig 11, which is the counterpart of Fig 4. Simulations shown in Fig 7 clearly indicate that predictions of peak-to-peak decline rates made by Fig 4 are quantitatively more reliable than those made by Fig 11.

The quantitative analysis presented in section 3.1 focuses on the rate of peak-to-peak decline, with the intent to design pulse dosing regimens for rapid bacterial eradication. A somewhat more accurate characterization of time to eradication would be provided by focusing on the rate of dip-to-dip decline (c.f. Figs 6 and 7). It can be shown (Eqs (SI-1) and (SI-2) in Appendix A in S1 Text), however, that both peaks and dips of the bacterial population follow the same rate, governed by the eigenvalues $\lambda_1$, $\lambda_2$ of the matrix **M** (Eq (6)). Because tracking peaks is simpler to analyze than tracking dips, the choice was made to focus on the former, leaving the latter for a future study.

In addition to the quantitative analysis that resulted in the selected values $t_{on}$ = 3h, $t_{off}$ = 2h (Eq (17)) for the pulse dosing experiment, heuristic analysis was also used to corroborate that choice, as follows:

In the early part of the constant dosing (control) experiment, it is evident (Fig 6) that around 3h persisters start becoming an increasingly significant fraction of the bacterial population (as manifest by the bend in the declining population logarithmic size) and subsequently dominate the population. Therefore, keeping the antibiotic on beyond 3h would result in reduced killing rate. Furthermore, while turning the antibiotic off starts driving persisters back to normal cells, keeping it off beyond 2h would reach the limits of that drive. While the above analysis suggests that the selected of $t_{on}$, $t_{off}$ values are sensible, it cannot provide a general characterization of optimal values or establish the importance of the ratio $t_{on}/t_{off}$.

Related experiments [33] have found that persisters treated for 3h with Ampicillin were able to resuscitate after growth in fresh media for as little as 1h. In our experiments also, a manifold increase in kill rates when treated again with Ampicillin suggested that persisters resuscitated to normal cells indeed within an hour of growth in fresh media.

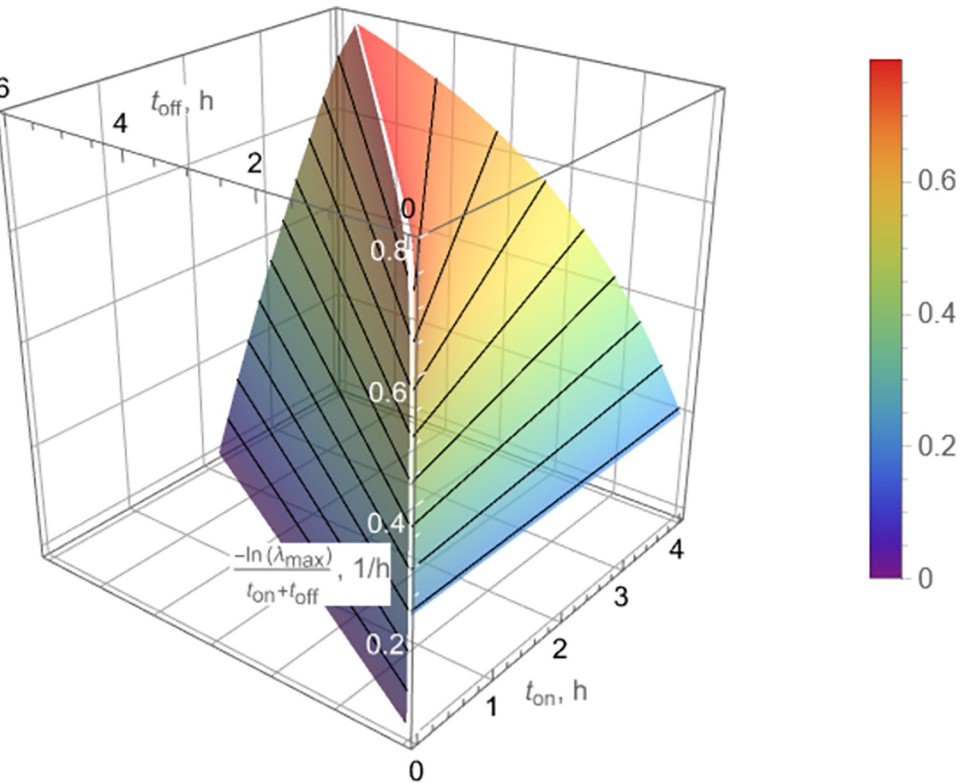

**Fig 11. Peak-to-peak actual-time decline rate corresponding to the larger (slower) of the two eigenvalues $\lambda_1$, $\lambda_2$ of the matrix M as a function of $t_{on}$, $t_{off}$ for the values of $K_{n,on}$, $K_{p,on}$, $K_{n,off}$, $K_{p,off}$ in Table 1.** The crease line (white), where the two surfaces intersect, corresponds to Eq (SI-16) and characterizes the highest peak-to-peak actual-time decline rate, for $t_{off}/t_{on} = 1.5$.

## 4.3 Modeling and parameter estimation

Estimation of the four model parameters in Eqs (1) and (2) faces the challenge that measurements of bacterial population size, Eq (4), are linear combinations of the two solution modes, $q_1 \exp(\rho_1 t) + q_2 \exp(\rho_2 t)$ (Appendix D in S1 Text) with

$$\rho_{1,2} = \frac{1}{2}\left( K_n + K_p \pm \sqrt{4ab + (K_n - K_p)^2} \right) \quad (19)$$

which makes it difficult to estimate all four $\{K_n, K_p, a, b\}$ with good accuracy from data. Nevertheless, assumptions based on fundamental knowledge can facilitate estimation. These assumptions are in accordance with relevant literature [9] and similar observations in our own experiments.

When the antibiotic is on, it follows that $b \approx 0$, because persisters remain dormant, and $a \gtrsim 0$, as some normal cells switch to persisters due to antibiotic stress. Estimation of a nonzero $a$ yielded a very small value, in agreement with the assumption $a \approx 0$, i.e. that persisters can be sourced to the initial pool of cells.

When there is no antibiotic and cells are grown in fresh media, it follows that $a \approx 0$, as the tendency of normal cells to become dormant is negligible during the early exponential phase (e.g. in less than 3h of growth), whereas $b > 0$, as persisters will be resuscitated shortly (e.g. within an hour) after inoculation in fresh media [13,33,34]. In addition, $K_{p,on} \approx -b$, because $\mu_{p,on} \approx 0$ and $k_{p,on} \approx 0$.

The different bactericidal outcomes produced by constant and pulse dosing are evident in Fig 6A, where the (one-standard-error) confidence bands towards the end of the experimental period of 13h are clearly separated, with pulse dosing yielding no detectable bacterial load and constant dosing leaving a load of about 20 log(CFU/mL) at 13h. In addition, Fig 6B shows the fitted model projection until the bacterial population outcome resulting from constant dosing reaches 0 at about 30h, long after pulse dosing has reached the same outcome in less than 12h.

## 4.4 Application of the proposed theory to literature data

To further test the ability of the proposed theory to predict the bactericidal efficacy of pulse dosing regimens, we analyzed experimental data from two studies in literature [28,29]. These studies presented pulse dosing regimens on bacterial populations with persister bacteria.

To perform the test, estimates of $K_{n,\mathrm{on}}$ (decline slope of the logarithmic population of *normal* cells under antibiotic exposure) were obtained from data presented in corresponding figures for both studies (Tables 2 and 3); and estimates of $K_{n,\mathrm{off}}$ (growth slope of the logarithmic population of *normal* cells in growth) for [29] were either obtained from a corresponding figure in [28] (Table 2) or estimated based on past experience (Table 3). The critical values of the ratio $t_{\mathrm{off}}/t_{\mathrm{on}}$ were subsequently calculated for both references from Eqs (7) and (11), respectively (Tables 2 and 3). These values were then compared to the actual values used in experiments presented in each reference cited. Corresponding predictions from this comparison ere consistent with data in all cases. Specifically, Table 2 indicates that bacterial populations declined when $t_{\mathrm{off}}/t_{\mathrm{on}}<(t_{\mathrm{off}}/t_{\mathrm{on}})_c$ (Eq (7)) and grew when $t_{\mathrm{off}}/t_{\mathrm{on}}>(t_{\mathrm{off}}/t_{\mathrm{on}})_c$. In fact, the peak-to-peak slopes shown in Fig 4 of [28] are in agreement with the discrepancy between the values of $t_{\mathrm{off}}/t_{\mathrm{on}}$ used and the critical or optimal values. Similarly, Table 3 indicates that $t_{\mathrm{off}}/t_{\mathrm{on}}$ in [29] is higher than the critical value $(t_{\mathrm{off}}/t_{\mathrm{on}})_c$ for all four strains focused on, in agreement with observed increasing persister percentages in the bacterial populations studied.

For completeness, the optimal ratios $(t_{\mathrm{off}}/t_{\mathrm{on}})_{\mathrm{opt}}$ are also shown in Tables 2 and 3.

## 5 Conclusions and future work

We have developed a methodology for systematic design of pulse dosing regimens that can eradicate persistent bacteria. The methodology relies on explicit formulas that make use of easily obtainable data from time-growth and time-kill experiments with a bacterial population exposed to antibiotics. Several extensions of this work can be pursued, including:

**Table 2. Literature data analysis to test proposed theory for pulse dosing regimen design.**

| Data source | [28] | | |
|---|---|---|---|
| **Bacteria Strain** | *S. aureus* | | |
| **Antibiotic** | Ofloxacin | | |
| $K_{n,\mathrm{on}}$ (h$^{-1}$) | −0.6 (Fig 4C) | | |
| $K_{n,\mathrm{off}}$ (h$^{-1}$) | 2.5 (Fig 4A) | | |
| $\left(\frac{t_{\mathrm{off}}}{t_{\mathrm{on}}}\right)_c = -\frac{K_{n,\mathrm{on}}}{K_{n,\mathrm{off}}}$ (Eq (7)) | 0.24 | | |
| $\frac{t_{\mathrm{off}}}{t_{\mathrm{on}}}$ used in [28] | 4h/20h = 0.2 | 9h/15h = 0.6 | 16h/8h = 2 |
| $\frac{t_{\mathrm{off}}}{t_{\mathrm{on}}}$ used in [28] $< \left(\frac{t_{\mathrm{off}}}{t_{\mathrm{on}}}\right)_c$ ? (Eq (7)) | Yes | No | No |
| Bacterial population decline? | Yes ([28] Fig 4B) | No ([28] Fig 4C) | No ([28] Fig 4D) |
| $\left(\frac{t_{\mathrm{off}}}{t_{\mathrm{on}}}\right)_{\mathrm{opt}} \approx \frac{\left(\frac{t_{\mathrm{off}}}{t_{\mathrm{on}}}\right)_c}{\left(\frac{t_{\mathrm{off}}}{t_{\mathrm{on}}}\right)_c+2}$ (Eqs (11), (14)) | 0.1 | | |

**Table 3. Literature data analysis to test proposed theory for pulse dosing regimen design.**

| Data source | [29] | | | |
|---|---|---|---|---|
| Bacteria Strain | *P. aeruginosa* | *A. baumannii* | *K. pneumoniae* | *E. aerogenes* |
| Antibiotic | Amikacin | | | |
| $K_{n,\text{on}}$ (h$^{-1}$) | −3 ([29] Fig 2A) | −3 ([29] Fig 2A) | −5 ([29] Fig 2A) | −5 ([29] Fig 2A) |
| $K_{n,\text{off}}$ (h$^{-1}$) | 1 to 3 (Estimate) | 1 to 3 (Estimate) | 1 to 3 (Estimate) | 1 to 3 (Estimate) |
| $\left(\frac{t_{\text{off}}}{t_{\text{on}}}\right)_c = -\frac{K_{n,\text{on}}}{K_{n,\text{off}}}$ (Eq (7)) | 1 to 3 | 1 to 3 | 1.7 to 5 | 1.7 to 5 |
| $\frac{t_{\text{off}}}{t_{\text{on}}}$ used in [29] | 43h/5h = 8.6 | 43h/5h = 8.6 | 43h/5h = 8.6 | 19h/5h = 3.8 |
| $\frac{t_{\text{off}}}{t_{\text{on}}}$ used in [29] $< \left(\frac{t_{\text{off}}}{t_{\text{on}}}\right)_c$ ? | No | No | No | No |
| Persister percentage decline? | No ([29] Fig 3a) | No ([29] Fig 3a) | No ([29] Fig 3a) | No ([29] Fig 3a) |
| $\left(\frac{t_{\text{off}}}{t_{\text{on}}}\right)_{\text{opt}} \approx \frac{\left(\frac{t_{\text{off}}}{t_{\text{on}}}\right)_c}{\left(\frac{t_{\text{off}}}{t_{\text{on}}}\right)_c + 2}$ (Eq (14)) | 0.3 to 0.6 | 0.3 to 0.6 | 0.5 to 0.7 | 0.5 to 0.7 |

- Test the outlined strategy on various pairs of pathogenic bacterial strains and antibiotics, including combinations of antibiotics for stubborn infections.

- Extend and test the developed methodology, both theoretically and experimentally, to

○ Clinically relevant pharmacokinetic profiles of antibiotic administration, e.g. periodic injection followed by exponential [35].

○ Reduced antibiotics concentration in consecutive cycles [27].

○ Ultimately in vivo studies

- The experiments presented used Ampicillin (a $\beta$-lactam antibiotic) which is a time-dependent antibiotic, namely it exhibits best efficacy if administered in periodic injections of as high concentration as possible. It is worth exploring the performance of the proposed pulse dosing design methodology to concentration-dependent antibiotics such as aminoglycosides and quinolones [36].

- Models developed using data from flow cytometry experiments [33] can better monitor the heterogeneity of a bacterial population, with potential improvements in modeling and, as a result, pulse dosing design.

- Extension to viable but not culturable cells: In addition to persisters, a well-known bacterial phenotype that can survive exposure to antibiotics is viable but not culturable (VBNC) cells [19, 37, 38]. The pulse dosing methodology presented here can, in principle, be applied to such cells, as they can be restored to normal growth and susceptibility to antibiotics upon provision of appropriate stimuli [39, 40]. However, experimentally studying this case poses different challenges, as standard cell cultures cannot be routinely used, and this is left for future exploration.

## Supporting information

**S1 Text. Appendix A.** Derivation of Eq (7); **Appendix B.** Optimal rate of decline for bacterial population peaks characterized by Eq (9); **Appendix C.** Estimation of $K_{n,\text{off}}$, $K_{n,\text{on}}$ from data in Fig 5; **Appendix D.** Analytical solution of Eqs (1) and (2).
(DOCX)

## Acknowledgments

MN and GS gratefully acknowledge substantial help provided by Sayed Golam Mohiuddin to set up and conduct the experimental part of the study.

## Author Contributions

**Conceptualization:** Garima Singh, Mehmet A. Orman, Jacinta C. Conrad, Michael Nikolaou.

**Data curation:** Garima Singh, Michael Nikolaou.

**Formal analysis:** Garima Singh, Jacinta C. Conrad, Michael Nikolaou.

**Funding acquisition:** Jacinta C. Conrad, Michael Nikolaou.

**Investigation:** Garima Singh, Michael Nikolaou.

**Methodology:** Garima Singh, Mehmet A. Orman, Jacinta C. Conrad, Michael Nikolaou.

**Project administration:** Jacinta C. Conrad, Michael Nikolaou.

**Resources:** Garima Singh, Mehmet A. Orman, Jacinta C. Conrad, Michael Nikolaou.

**Software:** Garima Singh, Michael Nikolaou.

**Supervision:** Jacinta C. Conrad, Michael Nikolaou.

**Validation:** Garima Singh, Mehmet A. Orman, Michael Nikolaou.

**Visualization:** Garima Singh, Jacinta C. Conrad, Michael Nikolaou.

**Writing – original draft:** Garima Singh.

**Writing – review & editing:** Garima Singh, Mehmet A. Orman, Jacinta C. Conrad, Michael Nikolaou.

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
