## [Decision Letter · Decision Letter 0]

26 Aug 2022

Dear Prof Nikolaou,

Thank you very much for submitting your manuscript "Systematic Design of Pulse Dosing to Eradicate Persister Bacteria" for consideration at PLOS Computational Biology.

As with all papers reviewed by the journal, your manuscript was reviewed by members of the editorial board and by several independent reviewers. In light of the reviews (below this email), we would like to invite the resubmission of a significantly-revised version that takes into account the reviewers' comments.

In particular, the reviewers raise important questions about features of, and apparent discrepancies in the data; these need to be addressed. Also, two reviewers remark that text and references in the manuscript do not acknowledge recent progress in the field. Please update the Introduction and Discussion sections accordingly. In addition, multiple reviewers complain about the design of the figures.

We cannot make any decision about publication until we have seen the revised manuscript and your response to the reviewers' comments. Your revised manuscript is also likely to be sent to reviewers for further evaluation.

Sincerely,

Rutger Hermsen

Guest Editor

PLOS Computational Biology

Rob De Boer

Section Editor

PLOS Computational Biology

Reviewer's Responses to Questions

**Comments to the Authors:**

Reviewer #1: This is an interesting study providing a computational model for eradicating dormant persister cells by pulse-dosing an antibiotic, and testing theoretical predictions. This reviewer only analyzed the experimental part of the study. The overall agreement between model and experiment is impressive. Specific comments follow.

1. “In fact, the mechanisms governing persister formation, survival, and resuscitation are diverse and not sufficiently understood (Balaban et al., 2013).”

- That was indeed true a decade ago. Since then, understanding of mechanisms governing persister formation improved significantly. See, for example, (Lewis, K., ed., (2019) Persister Cells and Infectious Disease, p. 295. Springer Nature Switzerland AG.)

2. “Such development can be broadly classified into two categories: (a) developing new anti-persister drugs, and (b) manipulating the dosing regimen of approved antibiotics.”

- The most common approach is combining approved antibiotics.

3. Fig. 5, treatment with constant administration of ampicillin over 3 hours results in an approximately 3log decrease in viable cells. Fig. 6, an experiment apparently identically designed, constant administration of ampicillin over 3 hours results in an approximately 5log decrease in viable cell. This discrepancy needs to be addressed.

Reviewer #2: Singh et al proposed a novel theoretical framework to predict the efficacy of pulsed antibiotic dosing in eradicating persister bacteria. I believe that the study is well conceived and it is backed up by a few experimental data. In my opinion the authors will need to address the issues reported below before this manuscript can be published in PLoS Computational Biology.

Abstract

I suggest rewrite/clarifying the following statement: “A key outcome of the theoretical analysis, on which the proposed methodology is based, is that dosing effectiveness depends mainly on the ratio of corresponding pulse durations.” It is not clear what the authors mean with ratio of pulse durations

Introduction

Lines 28-29. The field has moved forward from the notion that persisters must be in a slow growth state. A number of studies (including by one of the authors of the paper) suggest that persisters can also be in a growing state, e.g.:

doi:10.1128/AAC.00243-13

https://doi.org/10.7554/eLife.74062

https://www.embopress.org/doi/full/10.15252/msb.20145949

The authors should acknowledge these recent findings and amend this statement accordingly.

Lines 40-42: the authors correctly state that “our knowledge of persisters accelerated in the last two decades” but report nothing about most recent findings, e.g.:

https://www.science.org/doi/full/10.1126/science.aaf4762

https://www.sciencedirect.com/science/article/pii/S2211124722001516

https://journals.asm.org/doi/10.1128/mBio.00909-21

https://www.nature.com/articles/s41467-022-28141-x

https://journals.asm.org/doi/full/10.1128/mBio.00703-21

https://pubs.acs.org/doi/10.1021/acsinfecdis.1c00154

The authors could also add another phenotype, i.e. viable but non culturable cells, i.e.

doi:10.1128/AAC.00243-13

https://doi.org/10.1186/s12915-017-0465-4

10.1016/j.tim.2014.09.004

The framework they have developed could also help in the eradication of this surviving phenotype

Figure 2: it is not clear what it is being plotted on the third dimension (i.e. the vertical one that goes from -40 to 20)

Figure 5: the authors should clarify both in the figure and text which antibiotic is being used and what is its concentration. Why the data in Fig. 5B do not display a biphasic trend typically indicating the presence of persisters? This might have a strong impact on the rest of the study since the authors use this data to extrapolate kon. Could the authors please clarify this important issue.

Lines 219-223: could the authors add statistical analysis to conclude that indeed the two different treatment regimes lead to statistically different outcomes? It looks like there are only 10 CFU/ml in the case of constant dosing

Section 3.4.2 and Figure 8: I found these data confusing since in this case pulse dosing looks far worse than constant dosing. Is this because the pulses are so short? The authors should better clarify this in the manuscript and perhaps compare different pulse dosing regimes next to each other one with advantageous and one with disadvantageous outcome

Discussion

In its present state the discussion reads more as a summary than putting this new manuscript in the context of the persister research field. The authors could compare their findings with existing data in the literature, including the studies suggested above.

Reviewer #3: The authors used a simple mathematical model to study the optimal pulsing strategy to apply antibiotics and eliminate persister bacterial cells.

I believe this is a valid and interesting contribution to the field.

My main concerns are:

1- the type of plots used in figures 2, 3, 4 and 11 are difficult to interpret. I would check if series of 2-dimensional plots could be more easily interpretable than the current plots.

2- The authors used in vitro experimental data to fit parameter values. If a is 0 both in in the antibiotic on and off states, how are the persister cells generated?

3- As the model parameters can change with bacterial species and with the type of antibiotic, it would be useful to present a sensitivity analysis showing how the peak to peak decay rate varies with the different parameters.

4- The model describes well the decay of bacterial populations in vitro. The authors should discuss what is different with the population dynamics in vivo (during an infection), and if the model needs to be adapted in that scenario.

**Have the authors made all data and (if applicable) computational code underlying the findings in their manuscript fully available?**

Reviewer #1: Yes

Reviewer #2: Yes

Reviewer #3: **No: **The authors should make their mathematica and matlab code publicly available.

PLOS authors have the option to publish the peer review history of their article (what does this mean?). If published, this will include your full peer review and any attached files.

Reviewer #1: **Yes: **Kim Lewis

Reviewer #2: No

Reviewer #3: No
---

## [Decision Letter · Decision Letter 1]

29 Nov 2022

Dear Prof Nikolaou,

We are pleased to inform you that your manuscript 'Systematic Design of Pulse Dosing to Eradicate Persister Bacteria' has been provisionally accepted for publication in PLOS Computational Biology.

Best regards,

Rutger Hermsen

Guest Editor

PLOS Computational Biology

Rob De Boer

Section Editor

PLOS Computational Biology

Reviewer's Responses to Questions

**Comments to the Authors:**

Reviewer #1: the authors adequately addressed critiques

Reviewer #2: The authors have now addressed all the issues raised and the manuscript is ready for publication.

Reviewer #3: The authors have addressed all my previous comments.

**Have the authors made all data and (if applicable) computational code underlying the findings in their manuscript fully available?**

Reviewer #1: Yes

Reviewer #2: Yes

Reviewer #3: Yes

PLOS authors have the option to publish the peer review history of their article (what does this mean?). If published, this will include your full peer review and any attached files.

Reviewer #1: No

Reviewer #2: No

Reviewer #3: No

---

## [Editor Report · Acceptance letter]

11 Jan 2023

PCOMPBIOL-D-22-00793R1 

Systematic Design of Pulse Dosing to Eradicate Persister Bacteria

Dear Dr Nikolaou,

I am pleased to inform you that your manuscript has been formally accepted for publication in PLOS Computational Biology. Your manuscript is now with our production department and you will be notified of the publication date in due course.

With kind regards,

Anita Estes
